# In Vivo Investigation of (2-Hydroxypropyl)-β-cyclodextrin-Based Formulation of Spironolactone in Aqueous Solution for Paediatric Use

**DOI:** 10.3390/pharmaceutics14040780

**Published:** 2022-04-03

**Authors:** Antonio Lopalco, Annachiara Manni, Alexander Keeley, Shozeb Haider, Wenliang Li, Angela Lopedota, Cosimo Damiano Altomare, Nunzio Denora, Catherine Tuleu

**Affiliations:** 1Department of Pharmacy-Pharmaceutical Sciences, University of Bari “Aldo Moro”, 70125 Bari, Italy; antonio.lopalco@uniba.it (A.L.); angelaassunta.lopedota@uniba.it (A.L.); cosimodamiano.altomare@uniba.it (C.D.A.); 2School of Pharmacy, University College of London, 29/39 Brunswick Square, London WC1N 1AX, UK; annachiara.manni@studenti.unipr.it (A.M.); a-keeley@hotmail.co.uk (A.K.); shozeb.haider@ucl.ac.uk (S.H.); c.tuleu@ucl.ac.uk (C.T.); 3Food and Drug Department, University of Parma, Parco Area Delle Scienze 27/A, 43124 Parma, Italy; 4Imperial College London, South Kensington Campus, London SW7 2AZ, UK; wenliang.li@cranfield.ac.uk; 5Cranfield Water Science Institute, School of Water, Environment and Energy, Cranfield University, Cranfield MK43 0AL, UK

**Keywords:** spironolactone, hydroxypropyl-β-cyclodextrin, palatability, cyclodextrin inclusion, phase solubility study, brief-access taste aversion, human taste panel, paediatric formulation

## Abstract

Spironolactone (SPL), a potent anti-aldosterone steroidal drug used to treat several diseases in paediatric patients (e.g., hypertension, primary aldosteronism, Bartter’s syndrome, and congestive heart failure), is not available in child-friendly dosage forms, and spironolactone liquids have been reported to be unpalatable. Aiming to enhance SPL solubility in aqueous solution and overcome palatability, herein, the effects of (2-hydroxypropyl)-β-cyclodextrin (HP-β-CyD) were thoroughly investigated on solubilisation in water and on masking the unpleasant taste of SPL in vivo. Although the complexation of SPL with HP-β-CyD was demonstrated through phase solubility studies, Job’s plot, NMR and computational docking studies, our in vivo tests did not show significant effects on taste aversion. Our findings, on the one hand, suggest that the formation of an inclusion complex of SPL with HP-β-CyD itself is not necessarily a good indicator for an acceptable degree of palatability, whereas, on the other hand, they constitute the basis for investigating other cyclodextrin-based formulations of the poorly water-soluble steroidal drug, including solid dosage forms, such as spray-dried powders and orodispersible tablets.

## 1. Introduction

Spironolactone (SPL) is a semi-synthetic steroidal drug that exerts diuretic activity by competing for cytoplasmic aldosterone receptors and antiandrogen activity by competitive binding to the androgen receptor [1]. SPL is currently used for the treatment of several diseases in children, including hypertension, primary aldosteronism, Bartter’s syndrome and congestive heart failure [2,3,4,5,6].

SPL is classified as a class II drug according to the Biopharmaceutics Classification System (BCS); it has poor solubility in water (~24 μg/mL), and the bioavailability of SPL preparations is known to vary significantly among brands and batches. SPL has been marketed in tablet form (25 mg, 50 mg, and 100 mg) for over 50 years. In 2017, it became available in some countries as a commercial oral suspension named CaroSpir^®^ (5 mg/mL). However, on top of only being licensed for use in adults, solid dosage forms are unsuitable for neonates, infants and children, who are not able to swallow them. As for liquid dosage forms, ensuring the use of safe excipients and concentrations leading to manageable dose volumes is necessary for children yet not central to the design of liquids for adults.

To overcome this shortcoming, pharmacists have traditionally extemporaneously prepared and personalised these liquids to allow SPL paediatric doses ranging from 1 to 3 mg/kg/day. In order to adjust the liquid’s strength for paediatric use, SPL tablets are usually crushed and dispersed in water [7]. The extemporaneous preparation of such oral suspensions may often represent the only alternative for the paediatric population to meet their therapeutic needs. However, even though they allow some major administration issues (e.g., dose flexibility and ease of administration) to be overcome, their formulation is quite challenging to prepare due to the heightened risk of physical instability of the suspensions and, compared to solutions, difficulties in administration because of the potential for dose withdrawal heterogeneity. Issues with solubility and bioavailability, unpleasant taste and the uniformity of the dose to ensure accurate and reproducible dosing must indeed be overcome [8].

Different guidelines have been proposed to ensure better quality of the prepared products, and a dedicated database has been built and shared to present stability data [9]. On the contrary, palatability is often not demonstrated for extemporaneous oral preparations used in clinical practice. In a recent patent, Pipho and De Hart reported some pharmaceutical compositions of palatable liquid formulations of SPL containing flavouring and sweetening agents to mask its unpleasant taste [10]. Palatability is a crucial aspect of oral liquid formulations, and it is a potential tool that can be used to improve patient acceptability and overall adherence [11,12]. In fact, the poor palatability of drugs and their formulations is central to non-compliance among children [13,14], with potential detrimental impact on the safety and efficacy of the therapy [15].

In light of the above, the search for appropriate taste-masking technologies has become pivotal in the pharmaceutical development of paediatric medicines. Several taste-masking strategies are available to enhance unpalatable active pharmaceutical ingredients (APIs) [16,17,18,19,20]. Among them, complexation with cyclodextrins (CyDs), which can be found in many marketed pharmaceutical products for various purposes [21], has been successfully used in human volunteers to mask the unpleasant taste of different APIs [22,23,24,25,26,27,28,29,30,31,32,33,34,35]. More so for solubilisation and bioavailability enhancement goals, CyDs have been applied in various pre-formulation and formulation studies with SPL. In fact, hydrophobic molecules, such as SPL in this case, with a size that fits into the relatively lipophilic inner CyD cavity, are able to replace water molecules and form a water-soluble inclusion complex in an aqueous solution [36,37,38]. Moreover, in vivo studies have indicated that CyD formulations enhance SPL bioavailability and that the derivatives of β-CyD could be safe and suitable excipients for the solubilisation of SPL in paediatric formulations [39,40,41], but none validated the taste enhancement provided.

The taste-masking mechanism of CyDs is also based on the complexation method. This method is particularly convenient for the excellent safety profile of orally administered CyDs [36], excipient toxicity being another key, yet not trivial, consideration when formulating for children. As a consequence of this inclusion, CyDs should be capable of masking the unpleasant taste of the drug by decreasing the interaction of API with taste buds, where taste receptors are located [42,43,44,45]. Although the inclusion complex formation is regarded as necessary to ensure the taste-masking efficacy of CyDs in a given drug, is it sufficient? Accordingly, appropriate taste evaluation methods should be considered to determine the efficacy of this taste-masking technique [46]. The gold standard method to establish the overall palatability is the human taste panel test, in which a group of healthy adults undertake a gustatory evaluation test. However, it should be taken into account that its use is restricted by logistics, monetary, ethical and safety concerns, especially during the preclinical or early clinical phases of drug development. Therefore, several non-human in vitro and in vivo taste evaluation methods have been developed to overcome these obstacles, such as the use of electronic tongues and the brief-access taste aversion (BATA) model. Specifically, the latter proved to be an efficient in vivo taste assessment tool for APIs dissolved in water, showing good correlations with human taste data [47].

The aim of the present work was to investigate the effect of a CyD-based solution of SPL on water solubilisation and taste-masking for paediatric formulations that could be produced extemporaneously in pharmacy dispensaries. Herein, the complexation of SPL with 2-hydroxypropyl-beta-cyclodextrin (HP-β-CyD) was proved by UV spectrophotometric and HPLC methods. Nuclear magnetic resonance (NMR) spectroscopy and molecular modelling studies were also performed in order to better understand the interactions between HP-β-CyD and SPL. An in vivo BATA model and a human panel test were the two taste assessment tools used to directly measure the taste-masking efficacy of HP-β-CyD in a clinically relevant 1 mg/mL SPL solution.

## 2. Materials and Methods

### 2.1. Materials

For the UV/Vis spectroscopy method, Job’s plot and NMR studies, SPL was supplied by Farmalabor SRL (Canosa di Puglia, BT), and HP-β-CyD MW ~1460 was purchased from Sigma-Aldrich Chemie GmbH (Steinheim, Germany).

For the HPLC studies, HP-β-CyD was kindly donated by FarmaLabor (Canosa di Puglia BT, Italy). Acetonitrile (ACN, HPLC grade) and purified water (Ph. Eur.) were purchased from Carlo Erba s.r.l. (Milan, Italy).

For the human taste panel, SPL was supplied by Fagron, UK, and KLEPTOSE^®^ HP-β-CyD Oral grade was supplied by Roquette, France.

The SPL and CyDs purchased from different suppliers had to be of pharmaceutical grade for human consumption in the sensory panel. Moreover, CyDs used in the experiments were characterised by the same degree of substitution (4.7).

### 2.2. Phase Solubility Studies

The phase solubility study of SPL in presence of HP-β-CyD was performed using UV and HPLC methods as described by Higuchi and Connors [48].

#### 2.2.1. UV/Vis Spectroscopy Method

An excess amount of SPL was added to five vials containing an aqueous solution (10 mL) of HP-β-CyD at 5 different concentrations (7.27 mM, 14.54 mM, 43.62 mM, 87.5 mM, 130.87 mM). The vials were sonicated for 15 min and shaken at a constant temperature (25 °C) to achieve equilibrium. After 72 h, the solutions were filtered through a nylon syringe filter (pore size: 0.45 μm), and the filtrates were diluted with deionised water. The solutions were then assayed by UV/Vis spectroscopy at 244 nm to determine SPL content using a Jenway 6305 UV/Vis spectrophotometer (Bibby Scientific, Staffordshire, UK). The concentration of each sample was determined using a calibration curve. A phase solubility diagram was constructed by plotting the dissolved SPL against the respective concentration of HP-β-CyD. Given that the stoichiometry of the complex is 1:1, the binding constant (*K*_1:1_) of the HP-β-CyD inclusion complexes with SPL (also found in the literature as association constant, stability constant, or complex constant) and the complexation efficiency (CE) were calculated using Equations (1) and (2):(1)$$K_{1:1}=\frac{slope}{S_{0}\left(1-slope\right)}$$
(2)$$CE=\frac{slope}{\left(1-slope\right)}$$
where slope is the slope of the obtained phase solubility diagram straight line, and $S_{0}$ represents the solubility of SPL in absence of HP-β-CyD.

#### 2.2.2. High-Performance Liquid Chromatography (HPLC) Method

Phase solubility analysis was performed in purified water containing 0–20% HP-β-CD with an excess of SPL. Each sample was equilibrated in an orbital shaking incubator at 25 °C at 50 rpm for 24 h. Then, they were centrifuged and filtered through a 0.22 µm pore size nylon membrane filter, and the supernatants were diluted before HPLC analysis.

An Agilent (Milan, Italy) 1260 Infinity Quaternary LC System equipped with an Agilent variable wavelength UV detector, a Rheodyne injector (Rheodyne, Model 7725i, Agilent) equipped with a 20 µL loop and OpenLAB CDS ChemStation software (Agilent) were used for HPLC analysis. A Zorbax Eclipse Plus C18 column (particle size 3.5 mm, 4.6 × 100 mm) thermostated at 20 °C was used. An isocratic separation was performed using purified water and ACN, 45/55 *v*/*v*, as the mobile phase at a flow rate of 1 mL/min. An injection volume of 20 µL was used, and the SPL was detected at 238 nm. The total acquisition time was 10 min.

Data fitting to determine *K*_1:1_ value was performed using GraphPad/Prism version 5.0 (GraphPad Software, La Jolla, CA, USA) using linear least-squares regression analysis. The stability constant of the complex (*K*_1:1_) and the complex efficiency (CE) were calculated using Equations (1) and (2).

### 2.3. Job’s Plot

To determine the stoichiometry of the SPL:HP-β-CyD inclusion complex, Job’s method was used: stock solutions of SPL were prepared with equimolecular concentration of the drug and HP-β-CyD. Different volumes of said solutions were mixed in such a way that the total concentration [HP-β-CyD] + [SPL] remained constant (0.05 mM) and that the molar fraction of SPL (X_SPL_ = [SPL]/([SPL] + [HP-β-CyD])) varied in the range 0–1. The samples were then sonicated for 15 min and shaken for 72 h at a constant temperature (25 °C) to reach equilibrium. The solutions were then suitably diluted and assayed by UV/Vis spectroscopy at 244 nm to determine SPL content using a Jenway 6305 UV/Vis spectrophotometer (Bibby Scientific, Staffordshire, UK). A graph was obtained by plotting ΔA in presence of the host with respect to the value of the free guest (ΔA × X_SPL_) vs. X_SPL_, where ΔA is the difference of the measured absorbance of SPL with and without HP-β-CyD. The value of X_SPL_ for which the plot presents the maximum deviation (r) gives the stoichiometry of the inclusion complex.

### 2.4. NMR Studies

In order to examine the complexation process between SPL and HP-β-CyD in solution, solutions of samples were prepared in 99.96% deuterated water (D_2_O) and dimethyl sulfoxide (DMSO-d_6_) provided by Cambridge Isotope Laboratories. First, ^1^H-NMR spectra of HP-β-CyD and SPL were recorded separately. Two stock solutions of 2.4 mM were prepared. Based on these two equimolar solutions, three samples containing both SPL and HP-β-CyD were prepared. This was achieved by mixing the two solutions at varying proportions so that the total concentration [HP-β-CyD] + [SPL] remained constant and X_SPL_ varied in the range 0–1. D_2_O was used to obtain HP-β-CyD and SPL:HP-β-CyD complexes spectra, whereas DMSO-d_6_ was used to obtain the SPL spectrum. Solution ^1^H spectra were recorded on a Bruker Avance 500 MHz NMR spectrometer equipped with a 5 mm cryoprobe. Data acquisition and processing were performed using standard TopSpin (version 3.2) software (Bruker, Billerica, MA, USA). NMR measurements were carried out at 298 K.

### 2.5. Molecular Modelling

The structure of HP-β-CyD was downloaded from the Protein Data Bank (code 5e6z), and the structure of SPL was sketched in LigEdit module implemented in the MolSoft ICM-Pro package [49]. SPL was then docked to the HP-β-CyD structure. A maximum of 30 docked conformations were generated, and, finally, visualisation of the docked poses was carried out using ICM Pro Molsoft molecular modelling package. The final conformations were chosen based on the interaction energy between HP-β-CyD and SPL. Visualisation of the docked poses was carried out using Pymol (Schrödinger) and ICM-Pro Molsoft molecular modelling package.

### 2.6. BATA Experiments

This model is a relatively simple and fast method that can successfully detect the aversive taste of APIs in an objective and quantitative manner. All the procedures were carried out in accordance with the United Kingdom (UK) Animals (Scientific Procedures) Act 1986 (Project Licence PPL 70/7668).

#### 2.6.1. Test Solutions

One BATA experiment aimed to assess the taste of HP-β-CyD alone at concentrations ranging between 2% and 25% *w/v*. The second one aimed to assess the taste-masking efficacy of HP-β-CyD on SPL 1 mg/mL. Six solutions containing SPL 1 mg/mL and HP-β-CD at different concentrations (from 0.85% to 18% *w/v*) were tested.

Furthermore, 1% (*w/v*) HP-β-CyD corresponded to the minimal calculated quantity of HP-β-CyD needed to prepare a solution of 1 mg/mL of SPL (as per equation presented in Figure 1a), and 0.85% was the minimum experimentally needed to obtain a concentration of SPL 1 mg/mL in soluion. The other percentages tested were to assess the added value of excess of HP-β-CyD on taste of SPL. Additionally, one saturated solution of SPL was prepared and tested as a negative (aversive) control. SPL was suspended in water 1 mg/mL, and this suspension was filtered (0.45 μm) to obtain a saturated SPL aqueous solution.

All the freshly prepared samples were magnetically stirred for 30–45 min and then sonicated for 15 min at room temperature (25 °C). The concentrations of HP-β-CyD and SPL in each formulation are summarised in Table 1. One sample of quinine hydrochloride (QHCl) was used at IC50 value (0.08 mM) as a negative bitter control.

#### 2.6.2. Animals

Ten midly water-deprived (22 h) male Sprague-Dawley (CD) rats (Charles-River, Kent, UK), aged approximatively 9 weeks at the beginning of the first experiment, were used in the present study. On arrival, rats were uniquely ear-punched for ease of identification, weighed and then housed in pairs in standard cages. They were housed in a room that was maintained at 21 ± 2 °C with 55 ± 10% humidity and with a 12:12 h light/dark cycle. All experiments occurred during the light phase of the cycle. Rats received 7 days of acclimatisation prior to any experiment, during which they had free access to food and tap water, and weight and water and food consumption were checked every day.

#### 2.6.3. Procedure

The BATA tests were conducted in a lickometer (Davis MS-160, DiLog instruments, Tallahassee, FL, USA), and each sample was presented to the rat via a small opening in the chamber of the lickometer. Access to stimuli was controlled by a computer in order to record the number of licks. Each rat underwent two testing days, during which they were randomly presented the samples (8 s each time) at least 3 times. Each sample was followed by a 2 s water rinse presentation. Between each presentation, a 5 s inter-presentation interval was observed.

#### 2.6.4. Data Analysis

Data are represented as notched boxplots indicative of the 95% confidence interval of the median. The distribution of the data was assessed with the Shapiro–Wilk test. Because the data were not normally distributed, non-parametric analysis was used. In particular, the Kruskal–Wallis test was performed to determine whether there was significant difference in the “number of licks” between the different concentrations tested. When significant, post hoc analysis using Xin Gao et al.’s non-parametric multiple test was subsequently performed to verify if each concentration was significantly different from the reference sample (e.g., water) at a significance level (α) of 0.05. Both statistical analyses and boxplots were generated using R software (open source).

### 2.7. Human Taste Panel

This study was approved by the UCL Research Ethics Committee (Project ID number: 4612/019).

#### 2.7.1. Taste Solutions

During the testing day, candidates assessed the following:One calibration sample (C): a saturated solution of SPL in water.Six test samples:⋅Formulation 1 (F1): SPL 1 mg/mL in 1% (*w/v*) HP-β-CyD and corresponding placebo (P1).⋅Formulation 2 (F2): SPL 1 mg/mL in 6% (*w/v*) HP-β-CyD and corresponding placebo (P2).⋅Formulation 3 (F3): SPL 1 mg/mL in 18% (*w/v*) HP-β-CyD and corresponding placebo (P3).

These solutions were previously assessed with BATA experiments. 

The samples were freshly prepared (as described in Section 2.6.1) under strict quality measures in a dedicated area under the supervision of a registered UK pharmacist, and they were presented at room temperature.

#### 2.7.2. Participants

A total of 24 healthy volunteers (11 females and 13 males) aged between 18 and 40 years old were recruited after screening for eligibility criteria. If they were a smoker, they had to forbear smoking at least one hour before and during all the tests. Any participant undergoing dental care or medicinal treatment (except contraceptives or over-the-counter medicines) during the 15 days before the test was unable to take part in the study.

#### 2.7.3. Procedure

The “swirl and spit” methodology was employed in this study. Throughout the session, the participants were presented with 10 mL samples labelled with a unique three-digit code in a randomised order. Each sample was assessed in duplicate. The participants had to rinse their mouths with the samples for 5 s to cover all oral surfaces and then spit the sample into a receptacle provided. As soon as the sample was spit out, they had to rate the taste using a computerised questionnaire with a 100 mm horizontal visual analogue scale (anchored from “not aversive” to “extremely aversive”). They were prompted to add descriptive comments on taste and smell (free text). Before and after each sample, participants rinsed their mouth with mineral water and could have unsalted crackers to neutralise their palate. Between each sample, an interval of up to 10 min was respected so that the previous sample was no longer perceived. Participants were allowed to immediately re-taste each sample once if needed. 

#### 2.7.4. Data Analysis

The same statistical tests described for the BATA data analysis were performed for the human taste panels. All data visualisations and statistical analyses were performed using R software (open source).

## 3. Results

### 3.1. Phase Solubility Studies

#### UV/Vis Spectroscopic and HPLC Analyses

The calibration curve of SPL, used to assay the concentration in the unknown samples, showed linearity over the concentration range of 6.6–52.8 mM, in accordance with Beer–Lambert’s law. The linear regression equation was y=0.0068−17.3946x, and the coefficient of determination (R^2^) was 0.998. Figure 1 shows the phase solubility plot for SPL as a function of HP-β-CyD. The equation for the best fit plot of SPL and HP-β-CyD, after linear regression analysis, was found to be [SPL] = 0.3901 [HP-β-CyD] − 0.5176, which showed that the solubility of SPL increased linearly as a function of HP-β-CyD (R^2^ = 0.9973). The phase solubility diagram indicates that HP-β-CyD is capable of forming a complex with SPL, and the solubility curve can be classified as type A_L_; therefore, the complex formation has an estimated molar ratio of 1:1. The apparent stability constant (K_1:1_) for the 1:1 inclusion complex and the complexation efficiency (CE) were calculated according to Equations (1) and (2), and their values were equal to 12,114 M^−1^ and 0.639, respectively. These values are in agreement with the results of previous studies [50]. Similar to the UV/Vis spectroscopy data, the HPLC data showed that the solubility of SPL significantly increased in the presence of HP-β-CyD. In fact, the solubility of the drug increased about 1000 fold in water in the presence of 20% of HP-β-CyD from 24.43 μg/mL (0.058 mM) to 22.85 mg/mL (54.86 mM). The linear increase in the solubility of SPL as a function of HP-β-CyD concentration indicated the formation of a water-soluble complex in the solution, with a calculated stability constant of 11,026 ± 0.141 M^−1^ and a complexation efficiency of 0.647 ± 0.08.

### 3.2. Job’s Plot

The 1:1 stoichiometry of the SPL:HP-β-CyD complex, suggested by the solubility experiments, was confirmed using the Job’s plot method. The change in the absorbance of the guest (SPL) during the addition of the host (HP-β-CyD), ΔA, was measured at 244 nm for the mixtures of SPL and HP-β-CyD in deionised water, with the total molarity remaining constant. A second-order polynomial fitting curve of ΔA as a function of XSPL = [SPL]/([SPL] + [HP-β-CyD]) is shown in Figure 1. Thus, Job’s plot demonstrates that this complex has a 1:1 stoichiometry because the maximum ΔA was seen at an X_SPL_ value of 0.5.

### 3.3. NMR and Molecular Modelling

In order to infer insights into the binding interactions between HP-β-CyD and SPL, NMR studies were carried out. The identification of HP-β-CyD protons was not determined due to the complexity of the spectrum. However, (i) signals between 4.91 and 5.25 ppm appearing as two sets of multiplets with centres of mass at 4.98 and 5.18 ppm are reasonably assigned to protons H1′ with (or without) 2-hydroxypropyl substituents in different positions; (ii) signals at 1.09 ppm can be most certainly assigned to the methyl protons H9′ of the hydroxypropyl substituents; (iii) signals at 3.8–3.9 ppm can be most likely attributed to the H3′ proton; and (iv) signals at 3.62 ppm can be attributed to the H7′ protons of HP-β-CyD (Figure 2). The presence of SPL in the mixture affected the H3′ signals of HP-β-CyD positioned in the inner surface of the molecule, as well as the H7′ signals located on the wider rim of the molecule, and larger chemical shift changes were observed for H7 and H4 of SPL. These upfield shifts are due to their proximity to the secondary hydroxyl groups of HP-β-CyD, suggesting the formation of a SPL:HP-β-CyD complex with a 1:1 stoichiometry mainly via the A ring of SPL.

In order to obtain further insight into the binding mode of SPL to HP-β-CyD, computational molecular docking studies were carried out, revealing an orientational preference of the drug in the cyclodextrin cavity in spite of the different initial configurations arbitrarily imposed. More specifically, five binding energies ranging from −18 to −16.91 kcal/mol were calculated to be the lowest, hence giving the five most stable conformations (Figure 3). According to the proposed model of the inclusion complex, SPL docks with its A ring in proximity to H3′, which is fully consistent with the NMR data, and it is not completely inside the cavity of HP-β-CyD due to the thioacetate group (–SCOCH_3_), which prevents interactions with the other rings of SPL in the toroidal cavity of HP-β-CyD and only allows the insertion of SPL within the lipophilic core of HP-β-CyD involving the α,β-unsaturated group present on the A ring (Figure 2b).

### 3.4. BATA Experiment

#### 3.4.1. Taste Assessment of HP-β-CyD

The taste of HP-β-CyD alone was assessed in rats in order to determine a concentration range that could be tested with the rat BATA model. Figure 4 shows that the number of licks decreased with the increase in the concentration of HP-β-CyD. This demonstrated that, at higher concentrations, HP-β-CyD was aversive to the rats. The number of licks recorded on both days for 6%, 18%, 21% and 25% HP-β-CyD was significantly different from that of deionised water (*p* < 0.05), but this was not the case for 2% and 12% HP-β-CyD. Overall, similar trends were obtained for each individual rat (data not shown). However, there was noticeable variability between the two testing days (Figure 4). On the first testing session, all the concentrations, except 2% and 6%, were significantly different from deionised water (*p* < 0.05), whereas on the second testing day, all the HP-β-CyD concentrations were as palatable as deionised water (*p* > 0.05). This could be due to an attenuation of the neophobic behaviour of the rats on the second exposure to the sample, as reported in previous studies [51].

From the graphs, it can be concluded that empty HP-β-CyD eliticed a taste that the rats did not like. It is hypothetised that cyclodextrins, as they had no guests, were possibly interacting with taste receptors, generating an unfamilar negative sensation not aligned with the often-cited mild sugary taste confered to cyclodextrins. Except for the three highest concentrations, the HP-β-CyD solutions tested did not elicit more than 50% of inhibition of licks over the two days.

#### 3.4.2. Taste-Masking Assessment of SPL with HP-β-CyD

Because the higher concentrations of HP-β-CyD elicited an aversive response in the first BATA experiment, HP-β-CyD solutions above 18% were not tested with SPL (Table 1). However, lower HP-β-CyD solutions of 0.85% and 1% were included, as they correspond, in theory, to just enough HP-β-CyD to solubilise SPL 1 mg/mL (see Section 2.6.1).

The concentration–response curve shown in Figure 5 indicates that the saturated SPL solution without HP-β-CyD was around 80% of the water lick pattern, so it was not very aversive to the rats. It also shows that the number of licks for SPL solutions (1 mg/mL) in different concentrations of HP-β-CyD increased slightly only for HP-β-CyD concentrations of 0.85% and 1% and then decreased as HP-β-CyD concentrations increased. Each concentration was significantly different from the water on testing day 1 (*p* < 0.05), whereas there were no significant differences on testing day 2 except for the inclusion of 0.85% and 12% concentrations.

Post hoc analysis also indicated that, on testing day 2, none of the samples were significantly different from the SPL-saturated solution (*p* > 0.05), whereas on testing day 1, only 12% and 18% solutions elicited a significantly lower number of licks (*p* > 0.05). This suggests that HP-β-CyD is not efficacious at taste masking SPL. Interday variability was also observed with 1 mg/mL SPL and HP-β-CyD, and it was more pronounced in solutions with a 2% concentration or above; this phenomenon could be associated with the exposition to a new tastant but, at present, especially cyclodextrins, which might trigger a neophobic response of the rats, leading to aversiveness to the testing solutions. As previously explained, this phenomenon could wear off when the tastant is presented a third time or if a different protocol (conditioning) is employed.

### 3.5. Human Taste Panel

A total of 24 participants were recruited, but only data from 23 participants (11 females and 12 males) were analysed due to 1 participant misusing the scales provided. As shown in Figure 6, the saturated SPL aqueous solution was rated as aversive (average ratings mid-scale) but not extremely, which aligns with the findings of the BATA experiments. The volunteers were able to differentiate between the aversiveness of the formulations and the placebos, as well as the aversiveness of the formulations at different concentrations of HP-β-CyD (*p* < 0.05). As expected, the formulations containing SPL were significantly more aversive than their corresponding placebos (*p* < 0.05), but the aversiveness decreased with the increase in the concentrations of HP-β-CyD, suggesting that higher concentrations of HP-β-CyD were able to reduce the aversiveness of SPL. However, no significant differences between the SPL formulations and the calibration samples were observed, except for the formulation with 1% HP-β-CyD concentration, which the volunteers reported to be significantly more aversive than the saturated SPL solution (0.024 mg/mL).

The results obtained are consistent with those obtained on the second day of testing in the BATA experiments. In fact, the rats’ preference increased for SPL solutions with a low HP-β-CyD concentration (1%, 2% and 6%), whereas the number of licks was lower for the SPL solution with a HP-β-CyD concentration of 18%. The SPL solution with the lowest concentration of HP-β-CyD (0.85%) was less palatable than the SPL solution without HP-β-CyD. Similarly, the participants showed a greater aversion to this sample and did not dislike the placebos.

Overall, considering the results of testing day 2 of the BATA experiment only and the human panel results, there is not a great taste-masking effect of HP-β-CyD on SPL.

## 4. Discussion

The inclusion complex formation between HP-β-CyD and SPL was investigated in an oral liquid formulation in order to evaluate the taste-masking efficacy of CyD over the drug in aqueous solution. According to earlier studies, the complexation with CyDs increases the aqueous solubility, dissolution rate and bioavailability of SPL [52,53,54], but, to date, it has not yet been established whether the complexation with HP-β-CyD may improve the palatability of SPL formulations.

The SPL:HP-β-CyD inclusion complexes in solution were firstly characterised by a phase solubility study and the continuous variation method (Job’s plot). In good agreement with previous works, the phase solubility study showed that the use of HP-β-CyD enhances the solubility of SPL as a consequence of the inclusion complex formation [55]. The phase solubility curve was classified as A_L_ type, indicating that the stoichiometry of the inclusion complex is 1:1, which was confirmed using the Job’s plot method. Very similar results regarding the apparent stability constant (*K*_1:1_) and the complexation efficiency (CE) were obtained using UV/Vis spectroscopy and HPLC methods, which resulted in *K*_1:1_ = 12,114 M^−1^ and CE = 0.639 for the former, and *K*_1:1_ = 11,026 ± 0.141 M^−1^ and CE = 0.647 ± 0.08 for the latter, meaning that only 64–65 out of 100 molecules of HP-β-CyD in solution were involved in the inclusion complex formation with SPL. A possible explanation for this value could be due to a shallow penetration of the drug into the cavity. In fact, the data obtained from ^1^H-NMR spectra and molecular docking calculations suggest an orientation of SPL into the HP-β-CyD cavity that achieves only weak drug/receptor intermolecular interactions. In the NMR spectrum of the SPL:HP-β-CyD mixture, the SPL protons that showed larger chemical shift changes were H4 (positioned in the lipophilic core of HP-β-CyD) and H7 (located in proximity to the secondary hydroxyl groups on the wider rim of HP-β-CyD), suggesting an inclusion complex formation mainly via the A ring of SPL. The five different orientations of SPL within the HP-β-CyD cavity that were detected with docking calculations indeed confirm this binding mode, showing that the thioacetate group of SPL lies on the secondary face of HP-β-CyD, thus impeding the drug in fully accommodating deeply into the CyD cavity.

The effect that the complexation, with this spatial guest/host fit, may have on the taste-masking efficiency of the HP-β-CyD-based liquid formulation compared with SPL alone was investigated in vivo using the rat BATA model and a human taste panel. Depending on the physicochemical properties of the API, in vitro taste assessment using an electronic tongue could be a viable option. However, a concentration-dependent signal needs to be obtained to ensure its usability. In the SPL case, the non-ionic character and poor aqueous solubility of the drug, which results in only minor effects on the membrane potential of sensors, provided too little conductivity; thus, this method was not deemed appropriate. Difficulties regarding the detection of other neutral compounds have previously been reported [56].

The data obtained in both in vivo experiments suggest that, with the increase in the concentration of HP-β-CyD in the formulation, there was little to no improvement in the taste of the SPL formulations compared to pure SPL. Interestingly, the rat BATA model proved to be an efficient taste assessment tool that can be used in similar studies alternatively to human panels. To date, the BATA model has not yet been extensively assessed for taste-masking strategies such as the use of CyDs. The testing protocol would need to be amended to alleviate the neophobic response observed and expand the usage repertoire of the BATA model going forward.

Several papers have been dedicated to the reduction or elimination of the unpleasant tastes of drugs by using CyD complexation techniques, and some pharmaceutical products have been developed and are on the market. This study aimed to investigate the effect of a CyD-based solution of SPL on water solubilisation and taste-masking for paediatric formulations that could also be prepared extemporaneously in a pharmacy. Approaches easily implementable in pharmacy practice could include the addition of sweeteners and possibly flavouring agents in order to improve the palatability of the SPL solution. This would not be optimal as a lean formulation, and it would have less safe excipients. In the future, further studies will be performed in order to investigate the CyD complexation effectiveness of SPL in the presence of substances such as hydrophilic polymers (i.e., HPMC), hydroxyl acids and surfactants to form multi-component ternary complexes. It is known that the use of these auxiliary substances gives a supramolecular stable ternary system in comparison to the binary system given by CyD drugs. This can be attributed to the synergistic effect of polymer and CyD solubilisation and effective taste-masking ability in the formation of stronger drug–CyD–water-soluble polymer ternary complexes.

Moreover, due to the high molecular weight and relatively low complexation efficiency of CyDs increasing the bulk formulation possibly beyond the accepted limits for oral administration, this approach could reduce not only the amount of CyDs but also the formulation cost [57].

Further studies could focus on the role of ternary components and the strength of the binding/stability constant [58,59] to cater for the solubility and taste-masking efficiency of SPL.

A larger body of work could be to retrospectively (literature-based exercise) but also prospectively (lab-based exercise) collect data on the binding constants of HP-b-CyD with other drugs where there is (ideally clinical) evidence of taste masking in order to explore or model this relationship. Moreover, the inclusion complex formation of HP-β-CyD with additional APIs should be assessed to corroborate the results found in the present study and to elucidate its role in the taste masking of other drugs.

## 5. Conclusions

There is urgent need to ease the administration and enhance the palatability of extemporaneously prepared oral dosage forms of SPL for paediatric use due to the lack of children-appropriate formulations on the market. Standard operating procedures and stability data are particularly important for hospital pharmacists who must guarantee both the quality and safety of the extemporaneous preparations dispensed. Taste masking is also crucial when designing SPL oral dosage forms specifically for children. Herein, a simple liquid formulation of SPL (1 mg/mL), with increasing concentrations of HP-β-CyD, was investigated in order to determine the effect of CyD on the drug’s solubility, ensuring easier reproducible dosing and enhanced palatability through unpleasant taste-masking effects. Our findings showed that, while increasing the solubility in an aqueous solution of SPL, the addition of HP-β-CyD, most likely because of weak drug/receptor interactions as inferred by NMR and molecular docking calculations, did not significantly improve the taste of the SPL CyD-based liquid formulation, as in vivo tests in rats and humans proved. These findings suggest that, when using CyDs to mask the unpleasant taste of a given API, the inclusion complex formation between the drug and CyD is a necessary but not sufficient condition affording taste-masking efficiency. While the herein investigated HP-β-CyD-containing solution did not show significant correction of SPL’s unpleasant taste, the physicochemical and computational results of this study may provide support for the design and preparation of other cyclodextrin-containing oral paediatric formulations, including solid dosage forms (e.g., spray-dried powders and orodispersible mini-tablets), based on other suitable taste-masking technologies.

## Figures and Tables

**Figure 1 pharmaceutics-14-00780-f001:**
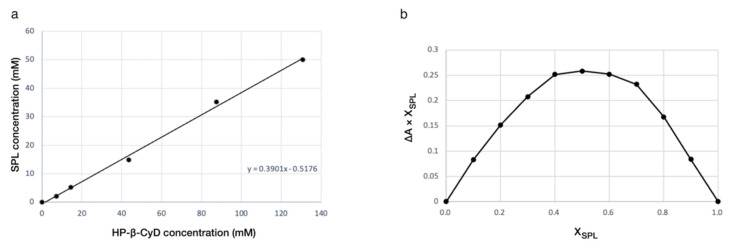
(**a**) Phase solubility diagram of SPL with HP-β-CyD in water at 25 °C; (**b**) Job’s Plot for 1:1 complex of SPL with HP-β-CyD in water at 25 °C.

**Figure 2 pharmaceutics-14-00780-f002:**
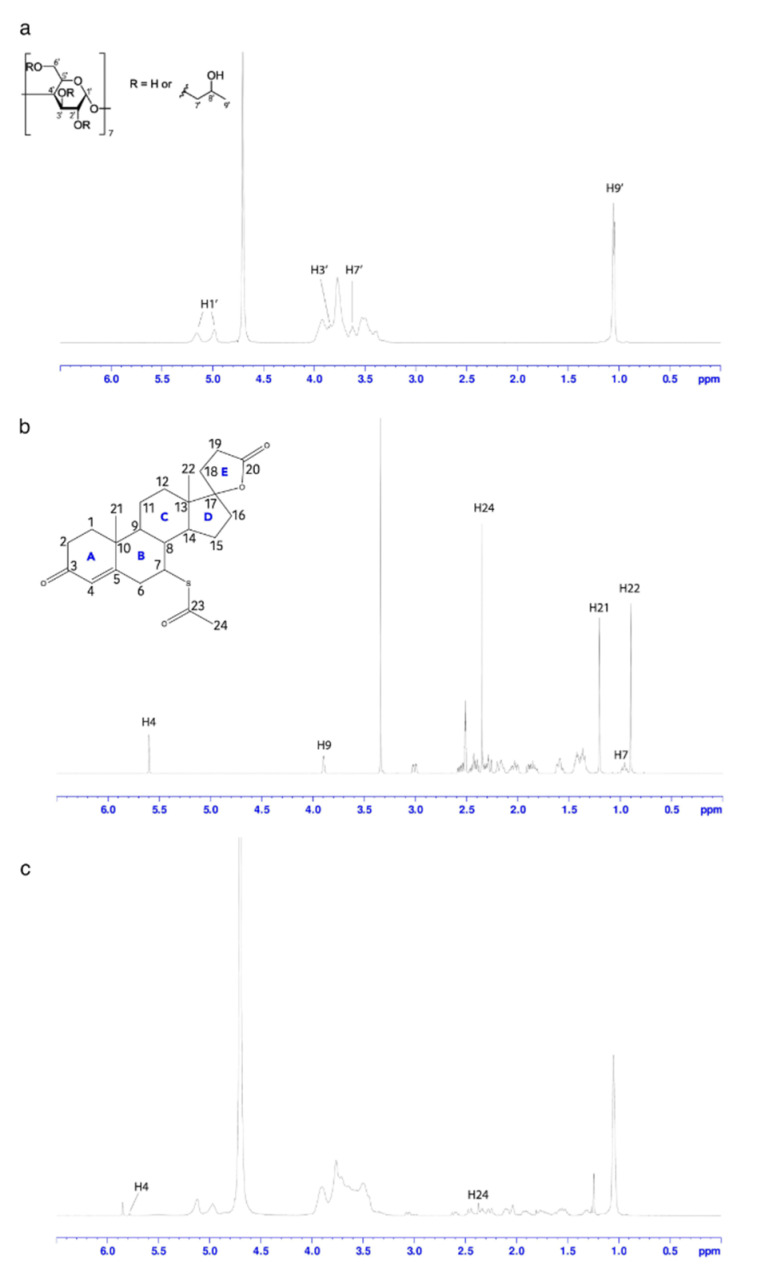
Chemical structure (with carbons numbered) and ^1^H NMR spectrum of (**a**) HP-β-CyD; (**b**) SPL; (**c**) SPL: HP-β-CyD mixture at 1:1 molar ratio.

**Figure 3 pharmaceutics-14-00780-f003:**
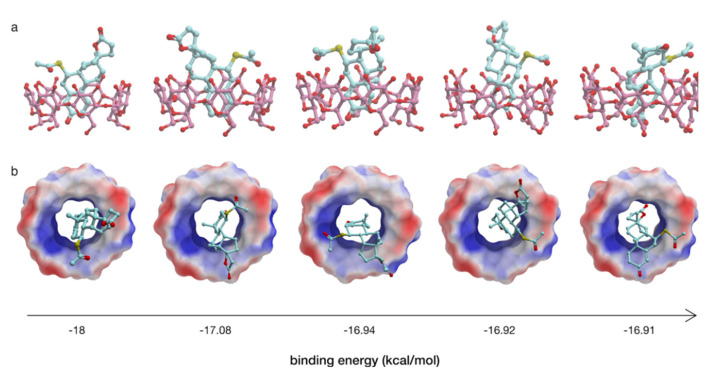
Docked conformations of SPL with HP-β-CyD. (**a**) Side view of the possible complexes formed between SPL (light blue) and HP-β-Cy (pink) and (**b**) top view of the complexes highlighting the electrostatic surface of HP-β-Cy. The binding energies (kcal/mol) are listed below the complexes.

**Figure 4 pharmaceutics-14-00780-f004:**
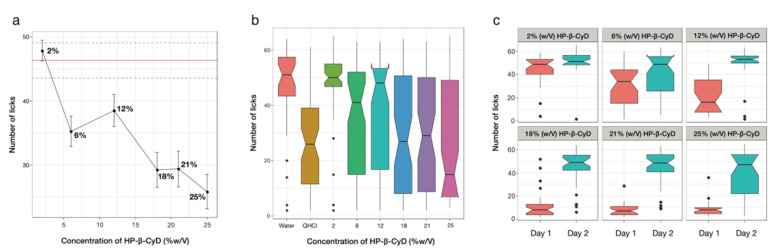
(**a**) Average number of licks (±SEM) as a function of concentration (% *w*/*v*) for HP-β-CyD, with the red lines representing the number of licks for water +/− 95% confidence interval; (**b**) boxplots representing the number of licks recorded for both testing days as a function of HP-β-CyD concentration (% *w*/*v*) − QHCl = quinine hydrochloride control at IC50; (**c**) boxplots representing the differences in number of licks recorded as a function of HP-β-CyD concentration (% *w*/*v*) between testing day 1 (red) and testing day 2 (blue).

**Figure 5 pharmaceutics-14-00780-f005:**
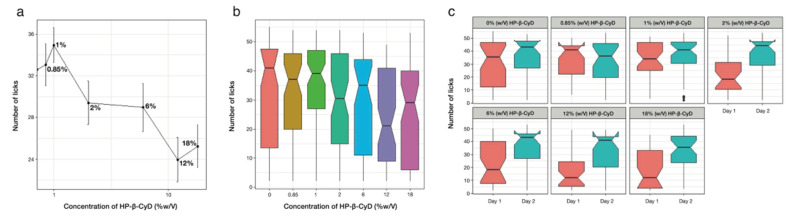
(**a**) Average number of licks (± SEM) as a function of HP-β-CyD concentration (% *w*/*v*) with 1 mg/mL SPL; (**b**) boxplots representing the number licks as a function of HP-β-CyD concentration (% *w*/*v*) with 1 mg/mL SPL; (**c**) boxplots representing the number of licks recorded as a function of HP-β-CyD concentration (% *w*/*v*) with 1 mg/mL SPL between testing day 1 (red) and testing day 2 (blue).

**Figure 6 pharmaceutics-14-00780-f006:**
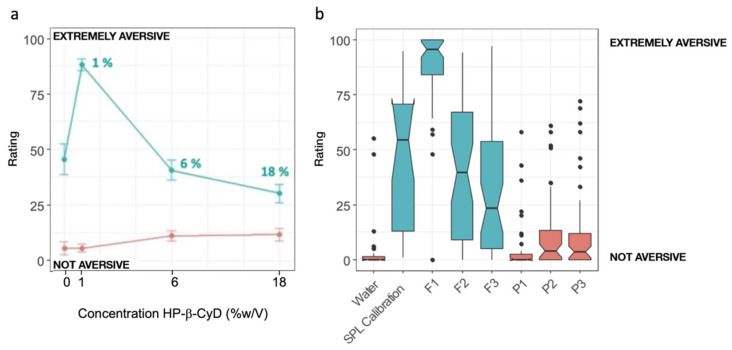
(**a**) Average aversiveness ratings (±the standard error of the mean, SEM) as a function of HP-β-CyD solutions with SPL (F1, F2, F3) in blue and their corresponding placebos (P1, P2, P3) in red; (**b**) boxplots representing taste ratings as a function of HP-β-CyD concentration for SPL solutions with HP-β-CyD and their corresponding placebos.

**Table 1 pharmaceutics-14-00780-t001:** Concentrations of HP-β-CD and SPL used in both BATA experiments.

Header	First BATA Experiment						Second BATA Experiment					
Sample	1	2	3	4	5	6	7	8	9	10	11	12
HP-β-CyD (*w/v*)	2%	6%	12%	18%	21%	25%	0.85%	1%	2%	6%	12%	18%
SPL (mg/mL)	−	−	−	−	−	−	1	1	1	1	1	1

## Data Availability

Not applicable.

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
