# Peer review of "In Vivo Investigation of (2-Hydroxypropyl)-β-cyclodextrin-Based Formulation of Spironolactone in Aqueous Solution for Paediatric Use"

_pharmaceutics, 2022, doi:10.3390/pharmaceutics14040780_

Round 1

Reviewer 1 Report

The manuscript titled “In vivo investigation on (2- hydroxypropyl)-β-cyclodextrin- based formulation of spironolactone in aqueous solution for pediatric use” does not appear to add significance to the current research space. Although the study design is pertinent, the formulation design/composition is inadequately chosen. Below are the concerns that should have been evaluated.

  1. Use of cyclodextrins as taste masking agent is widely known. However, the application of cyclodextrins for taste masking is commonly restricted to drugs that forms complex with high binding/stability constant because a low stability constant would lead to a rapid release of free drug in the oral cavity, resulting in inefficient taste masking. Mere use of cyclodextrins do not offer adequate taste masking.
  2. The authors should have evaluated methods of complex formation with higher stability constants to enable negligible amount of drug release in the oral cavity and offer a rapid release in the gastric lumen. Ternary complexation in the presence of HPMC, for example, has shown to be an effective method of increasing stability constant. This complexation could be utilized for effective taste masking by cyclodextrin.
  3. The authors have not proposed any further plan on re-formulating and re-valuation of other formulations for spironolactone.
  4. Moreover, there are several typographical errors in the manuscript. For example, “Cyclodestrin”, “arount” etc.

Author Response

The manuscript titled “In vivo investigation on (2- hydroxypropyl)-β-cyclodextrin- based formulation of spironolactone in aqueous solution for pediatric use” does not appear to add significance to the current research space. Although the study design is pertinent, the formulation design/composition is inadequately chosen. Below are the concerns that should have been evaluated.

  1. Use of cyclodextrins as taste masking agent is widely known. However, the application of cyclodextrins for taste masking is commonly restricted to drugs that forms complex with high binding/stability constant because a low stability constant would lead to a rapid release of free drug in the oral cavity, resulting in inefficient taste masking. Mere use of cyclodextrins do not offer adequate taste masking.
  2. The authors should have evaluated methods of complex formation with higher stability constants to enable negligible amount of drug release in the oral cavity and offer a rapid release in the gastric lumen. Ternary complexation in the presence of HPMC, for example, has shown to be an effective method of increasing stability constant. This complexation could be utilized for effective taste masking by cyclodextrin.
  3. The authors have not proposed any further plan on re-formulating and re-valuation of other formulations for spironolactone.

Thanks to the Reviewer for his/her comments.

The aim of our study was to investigate the effect on water solubilization and taste-masking of a CyD-based solution of SPL for liquid paediatric formulations, that could be prepared extemporaneously in a pharmacy. Our findings suggested that, while increasing the solubility in aqueous solution of SPL (a 1,000-fold increase of drug solubility in water was indeed achieved in the presence of 20% of HP-β-CyD), which is very important in terms of dosing uniformity, the addition of HP-β-CyD, most likely because of weak drug/CyD interactions (as inferred by NMR and molecular docking calculations), did not significantly improve the taste of the SPL CyD-based liquid formulation, as in vivo tests in rats and humans proved.

To better account for the aforementioned advantages and limitations, we complemented the Discussion as follows:

Several papers have been dedicated to the reduction or elimination of unpleasant taste of drugs by using CyD complexation techniques and some pharmaceutical products have been developed and are on the market. This study aimed at investigating the effect on water solubilization and taste-masking of a CyD-based solution of SPL for paediatric formulations, that could be also prepared extemporaneously in a pharmacy. Approaches easily implementable in the pharmacy practice could include the addition of sweeteners and possibly flavouring agents in order to improve the palatability of the SPL solution. This would not be optimal as a lean formulation with less and safe excipient. In the future, further studies will be performed in order to investigate the CyD complexation effectiveness of SPL in the presence of substances like hydrophilic polymers (i.e., HPMC), hydroxyl acids, and surfactants to form multi-component ternary complexes. It is known that the use of these auxiliary substances gives a supramolecular stable ternary system in comparison to the binary system CyD-drug. This can be attributed to the synergistic effect of polymer and CyD solubilization and effective taste masking ability on the formation of drug-CyD-water soluble polymer ternary stronger complexes.

Moreover, due to the high molecular weight and relatively low complexation efficiency of CyDs increasing the bulk formulation possibly beyond the accepted limits for oral administration, this approach could not only reduce the amount of CyDs but also the formulation cost [57].

Further studies could be focused on the role of ternary components and the strength of the binding/stability constant [58,59] to cater for the solubility and taste masking efficiency of SPL.”

4. Moreover, there are several typographical errors in the manuscript. For example, “Cyclodestrin”, “arount” etc.

Typographical errors in the manuscript have been corrected. Thank you for the suggestions.

Reviewer 2 Report

the manuscript describes a clinical relevant problem of a bad tasting drug that is used frequently as liquid in the pediatric population.

the methodology used to study the research questions are thorough and extensively described. Only one API has been studied, and the discussion might elaborate a bit more on how to intepretate these data for other APi'.s

results l.379; the bata rat model showed inconsistent results in the 2 days sessions. please elaborate on the conclusion on l. 491 that despite these results the rat model is considered an efficient tool. 

Author Response

The manuscript describes a clinical relevant problem of a bad tasting drug that is used frequently as liquid in the pediatric population.

the methodology used to study the research questions are thorough and extensively described. Only one API has been studied, and the discussion might elaborate a bit more on how to intepretate these data for other API's results l.379;

Thanks to the Reviewer for his/her comments.

This study was driven by a real clinical need and hence focused of one API only, that is spironolactone. The other drive was a relatively low-tech process enabling its extemporaneous preparation in a community pharmacy setting. The binding of one type of CD with one API has to be explored on a case-by-case basis. Yet we allude to a larger body of work that could be done, retrospectively but also prospectively, collect data on binding constants of HP-b-CyD (for example as it is the one we used) with other drugs where there is ideally clinical evidence of taste masking to explore or model this relationship.

the BATA rat model showed inconsistent results in the 2 days sessions. Please elaborate on the conclusion on l. 491 that despite these results the rat model is considered an efficient tool. 

Our work demonstrates that there may be differences in human and rat tolerance to HP-β-CyD on initial exposure, yet the findings are aligned and would have been indicative of the lack of taste masking of the CyD without the results of the human panel and access to another in vitro taste assessment such as an e-tongue. The BATA model offers good correlation to humans for many bitterness API but has not yet been extensively assessed for taste-masking strategies such as cyclodextrins. The model still needs to be further evaluated but pre-exposing the animals to the excipients prior to assessing them in formulated product could likely alleviate the neophobic response and allow for animals to assess HP-β-CyD containing formulations going forward (ongoing work).

Discussion has been added (Lines 493-497) in the manuscript:

“Interestingly, the rat BATA model proved to be an efficient taste assessment tool to be used in similar studies, alternatively to human panels during preclinical phases. BATA model has not yet been extensively assessed for taste-masking strategies such as cyclodextrins. The testing protocol would need to be amended to alleviate the neophobic response observed and expand the usage repertoire of the BATA model going forward.”

Reviewer 3 Report

In this paper, the authors prepared SPL-CD inclusion complexes to enhance the active principle solubility in water and mask its unpleasant taste. The paper is really well-conducted and the analyses are accurate, therefore, in my opinion, it can be published after minor revisions:

  • In the introduction, the authors should mention the use of micronizing techniques to obtain solid forms with cyclodextrins. For example, supercritical carbon dioxide-based techniques are frequently used for the complexation: doi: 10.1016/j.jcou.2020.101397; doi: 10.1007/s10847-019-00970-2
  • In the materials section, the authors used SPL purchased from Farmalabor for some analyses and from Fagron for the human taste panel. Are they sure that the compounds are identical?

Author Response

In this paper, the authors prepared SPL-CD inclusion complexes to enhance the active principle solubility in water and mask its unpleasant taste. The paper is really well-conducted and the analyses are accurate, therefore, in my opinion, it can be published after minor revisions:

  • In the introduction, the authors should mention the use of micronizing techniques to obtain solid forms with cyclodextrins. For example, supercritical carbon dioxide-based techniques are frequently used for the complexation: doi: 10.1016/j.jcou.2020.101397; doi: 10.1007/s10847-019-00970-2

Thanks to the Reviewer for the suggestion.

The aim of our study was to investigate the effect on water solubilization and taste-masking of a CyD-based solution of SPL for liquid paediatric formulations, that could be prepared extemporaneously even in a community pharmacy setting.

About micronizing techniques described in the articles suggested by this reviewer, we think that the supercritical carbon dioxide-based techniques are not appropriate for our goal, since these methods cannot be easily applied in a pharmacy or hospital pharmacy.

In the future, we will investigate these techniques for the complexation of the SPL and other drugs with CyDs and the production of solid dosage forms.

We really appreciated the suggestion and added these references in the revised manuscript.

  • In the materials section, the authors used SPL purchased from Farmalabor for some analyses and from Fagron for the human taste panel. Are they sure that the compounds are identical?

The SPL and CDs purchased from different suppliers had to be of pharmaceutical grade for human consumption in the sensory panel. Moreover, CDs used in the experiments were characterized by the same degree of substitution (4.7). This information has been added in the material section.

Reviewer 4 Report

The manuscript in vivo investigation on 2-hydroxypropyl Bcyclodextrin based formulation of spironolactone in aqueous solution for pediatric use provides an approach to mask the taste of spironolactone using cyclodextrin. The approach is valid but didn't show a significant improvement in taste masking. The author didn't offer any new insights into how to improve the binding. It is apparent that binding is weak with cyclodextrin  and the drug is not fully protected inside the cavity of the oligosaccharide. I suggest adding more information on how to improve the quality of binding into cylcodextrin. 

Author Response

The manuscript in vivo investigation on 2-hydroxypropyl Bcyclodextrin based formulation of spironolactone in aqueous solution for pediatric use provides an approach to mask the taste of spironolactone using cyclodextrin. The approach is valid but didn't show a significant improvement in taste masking. The author didn't offer any new insights into how to improve the binding. It is apparent that binding is weak with cyclodextrin and the drug is not fully protected inside the cavity of the oligosaccharide. I suggest adding more information on how to improve the quality of binding into cylcodextrin.

Thanks to this suggestion. Discussion has been improved in the manuscript with the following:

Several papers have been dedicated to the reduction or elimination of unpleasant taste of drugs by using CyD complexation techniques and some pharmaceutical products have been developed and are on the market. This study aimed at investigating the effect on water solubilization and taste-masking of a CyD-based solution of SPL for paediatric formulations, that could be also prepared extemporaneously in a pharmacy. Approaches easily implementable in the pharmacy practice could include the addition of sweeteners and possibly flavouring agents in order to improve the palatability of the SPL solution. This would not be optimal as a lean formulation with less and safe excipient. In the future, further studies will be performed in order to investigate the CyD complexation effectiveness of SPL in the presence of substances like hydrophilic polymers (i.e., HPMC), hydroxyl acids, and surfactants to form multi-component ternary complexes. It is known that the use of these auxiliary substances gives a supramolecular stable ternary system in comparison to the binary system CyD-drug. This can be attributed to the synergistic effect of polymer and CyD solubilization and effective taste masking ability on the formation of drug-CyD-water soluble polymer ternary stronger complexes.

Moreover, due to the high molecular weight and relatively low complexation efficiency of CyDs increasing the bulk formulation possibly beyond the accepted limits for oral administration, this approach could not only reduce the amount of CyDs but also the formulation cost [57].

Further studies could be focused on the role of ternary components and the strength of the binding/stability constant [58,59] to cater for the solubility and taste masking efficiency of SPL.”

Reviewer 5 Report

The manuscript presented a potentially helpful formulation of SPL for paediatric applications. Even though the results were unsatisfactory, the research method provided an idea for follow-up research. There are some issues to be settled before it can be accepted.

  1. The format and writing of the manuscript need to be checked carefully, e.g., 3.4. is missing.
  2. Introduction needs to be shortened.

Author Response

The manuscript presented a potentially helpful formulation of SPL for paediatric applications. Even though the results were unsatisfactory, the research method provided an idea for follow-up research. There are some issues to be settled before it can be accepted.

Thanks to the Reviewer for his/her useful comments.

  1. The format and writing of the manuscript need to be checked carefully, e.g., 3.4. is missing.

Thanks to this suggestion. The correction has been done in the manuscript.

  1. Introduction needs to be shortened.

Thanks to this suggestion. The introduction of the manuscript has been abridged.

Round 2

Reviewer 1 Report

Although the authors tried to address reviewer comments, these study results do not add any novelty or significance to the current research landscape on this topic.